# Loss of kallikrein-related peptidase 7 exacerbates amyloid pathology in Alzheimer's disease model mice

Kiwami Kidana[1,2,3,†], Takuya Tatebe[1,†], Kaori Ito[4], Norikazu Hara[5], Akiyoshi Kakita[6], Takashi Saito[7], Sho Takatori[1], Yasuyoshi Ouchi[2,8], Takeshi Ikeuchi[5], Mitsuhiro Makino[4], Takaomi C Saido[7] ⓘ, Masahiro Akishita[2], Takeshi Iwatsubo[9], Yukiko Hori[1] & Taisuke Tomita[1,*] ⓘ

## Abstract

Deposition of amyloid-β (Aβ) as senile plaques is one of the pathological hallmarks in the brains of Alzheimer's disease (AD) patients. In addition, glial activation has been found in AD brains, although the precise pathological role of astrocytes remains unclear. Here, we identified kallikrein-related peptidase 7 (KLK7) as an astrocyte-derived Aβ degrading enzyme. Expression of *KLK7* mRNA was significantly decreased in the brains of AD patients. Ablation of *Klk7* exacerbated the thioflavin S-positive Aβ pathology in AD model mice. The expression of Klk7 was upregulated by Aβ treatment in the primary astrocyte, suggesting that Klk7 is homeostatically modulated by Aβ-induced responses. Finally, we found that the Food and Drug Administration-approved anti-dementia drug memantine can increase the expression of *Klk7* and Aβ degradation activity specifically in the astrocytes. These data suggest that KLK7 is an important enzyme in the degradation and clearance of deposited Aβ species by astrocytes involved in the pathogenesis of AD.

**Keywords** Alzheimer's disease; amyloid-β; astrocyte; kallikrein-related peptidase 7; protease
**Subject Category** Neuroscience

## Introduction

Alzheimer's disease (AD) is the most common type of dementia. Genetic and biochemical evidence suggests that the aggregation and deposition of amyloid-β (Aβ) are critical processes in the pathogenesis of AD (Holtzman *et al*, 2011). Aβ is produced upon proteolysis of amyloid precursor protein (APP). Several genetic mutations linked to familial AD increase the production or aggregation of Aβ. In contrast, sporadic AD patients have been reported to have a decreased clearance rate, rather than an increased production rate of brain Aβ (Mawuenyega *et al*, 2010). Thus, understanding the molecular mechanism of brain "Aβ economy" (Karran *et al*, 2011), which reflects the balance of the rates of production, clearance, and aggregation of Aβ, is crucial for the development of effective therapeutics for AD. However, the whole picture of the pathophysiological Aβ clearance/degradation pathway in the brain is still unclear (Saido & Leissring, 2012). To date, several enzymes, including neprilysin (membrane metallo-endopeptidase) and insulin-degrading enzyme, have been identified as Aβ-degrading enzymes (Nalivaeva *et al*, 2014). Among them, the knockout of neprilysin or insulin-degrading enzyme in mice showed an approximately twofold increase in the level of soluble brain Aβ, although the effect on Aβ deposition *in vivo* remains controversial (Iwata *et al*, 2000, 2001; Farris *et al*, 2003; Leissring *et al*, 2003). Importantly, in a therapeutic context, remaining neurons might not be a suitable target to remove Aβ deposits in the treatment for AD. Thus, we focused on astrocytes, which play important roles in physiological and pathological functions in the brain (De Strooper & Karran, 2016; Pekny *et al*, 2016). Of note, changes in the phenotypes of astrocytes, but in not their number, were associated with the clinical pathology of AD

1 Laboratory of Neuropathology and Neuroscience, Graduate School of Pharmaceutical Sciences, The University of Tokyo, Tokyo, Japan
2 Department of Geriatric Medicine, Graduate School of Medicine, The University of Tokyo, Tokyo, Japan
3 Department of Internal Medicine, Komeikai Hospital, Tokyo, Japan
4 Venture Science Laboratories, R&D Division, Daiichi-Sankyo Co. Ltd., Tokyo, Japan
5 Department of Molecular Genetics, Brain Research Institute, Niigata University, Niigata, Japan
6 Department of Pathology, Brain Research Institute, Niigata University, Niigata, Japan
7 Laboratory for Proteolytic Neuroscience, RIKEN Brain Science Institute, Saitama, Japan
8 Federation of National Public Service Personnel Mutual Aid Associations, Toranomon Hospital, Tokyo, Japan
9 Department of Neuropathology, Graduate School of Medicine, The University of Tokyo, Tokyo, Japan
*Corresponding author. Tel: +81 3 5841 4868; E-mail: taisuke@mol.f.u-tokyo.ac.jp
†These authors contributed equally to this work

(Serrano-Pozo et al, 2013). However, the precise mechanism of astrocyte-mediated Aβ clearance remains unclear.

Fifteen kallikrein-related peptidase (KLK) family proteins have been identified and act in a complex network as a cascade reaction. Among them, KLK6 (neurosin) and KLK8 (neuropsin) are major kallikrein-related peptidase proteins in the central nervous system (Sotiropoulou et al, 2009; Prassas et al, 2015). KLK6 is widely expressed in several cells, and KLK8 is expressed in neurons. Also, KLK8 has been implicated in the pathogenesis of AD (Herring et al, 2016). KLK7 was originally identified as an inflammation-induced proteolytic enzyme in the skin. However, the expression level of KLK7 was decreased in the cerebrospinal fluid and brain of AD patients (Diamandis et al, 2004; Bossers et al, 2010). Moreover, it was reported that KLK7 is capable to cleave the hydrophobic core motif of Aβ fibrils, thereby attenuating neurotoxicity in vitro (Shropshire et al, 2014). In this manuscript, we have investigated the pathophysiological impact of KLK7 in brain Aβ economy and identified KLK7 as the astrocyte-derived Aβ-degrading enzyme that regulates amyloid pathology in vivo.

## Results

### Identification of KLK7 as the Aβ-degrading enzyme

To identify the proteolytic enzyme that is capable to degrade secreted Aβ species, we utilized the conditioned medium from 7PA2 cells as a substrate source, as 7PA2 cells secretes toxic human Aβ oligomer species (Podlisny et al, 1995). We took the conditioned medium from several cell lines and mixed with that from 7PA2 cells. The mixture was separated by Urea-containing SDS–PAGE gel, which enables us to discriminate different C-terminal length of Aβ immunoblot (Klafki et al, 1996; Qi-Takahara et al, 2005). We found that the media from astrocytoma CCF-STTG1, neuroglioma H4, neuroblastoma SH-SY5Y, and glioblastoma U87 cells showed robust Aβ-degrading activity in this condition (Fig 1A, and Appendix Fig S1A and B). In addition, the conditioned medium of CCF-STTG1 cells also showed the degrading activity on Aβ derived from human neuroblastoma BE(2)-C cells (Appendix Fig S1C). We then further characterized the degradation activity using CCF-STTG1 cells that showed strongest activity. This activity was specifically inhibited by diisopropyl fluorophosphates as well as tosyl phenylalanyl chloromethyl ketone (Fig 1B and C), indicating that a secreted chymotrypsin-type serine protease is required for Aβ degradation. Neither phosphoramidon, ethylenediaminetetraacetic acid nor GM6001 inhibited this activity, indicating that metalloprotease was not involved in this catabolic pathway (Fig 1B and D) (Iwata et al, 2000; Yin et al, 2006). However, inhibitors against known Aβ-degrading serine proteases [i.e., anti-plasmin for plasmin (Van Nostrand & Porter, 1999) and acetylmethionine for acyl-peptide hydrolase (Yamin et al, 2007)] failed to inhibit this activity (Fig 1E). In contrast, the activity was reduced by zinc ions (Fig 1F). This unusual character led us to investigate the possible function of KLK7, one of the secreted chymotrypsin-type serine proteases with zinc sensitivity (Debela et al, 2007).

It was reported that KLK7 is capable to cleave the hydrophobic core motif of Aβ fibrils in vitro (Shropshire et al, 2014). Moreover, the expression level of KLK7 was decreased in the cerebrospinal fluid and brain of AD patients (Diamandis et al, 2004; Bossers et al,

2010). However, it remains unclear whether KLK7 is involved in the brain amyloid pathology in vivo. We confirmed the ~50% reduction of KLK7 mRNA expression in the brains of Japanese AD patients (Fig 2) (Miyashita et al, 2014) in good correlation with Braak NFT stage (Appendix Fig S2). In addition, we analyzed the levels of human KLK7 mRNA expression in two public RNAseq datasets deposited at the AMP-AD knowledge portal: the Mayo RNAseq (MayoRNAseq) (Allen et al, 2016) and Mount Sinai Brain Bank (MSBB) AD cohorts. In the Mayo sample set, KLK7 repression was significantly decreased in the temporal cortex of AD patients (false discovery rate (FDR) < 0.05, β = −0.623). Similarly, KLK7 expression was significantly reduced in relation to increased amyloid plaque burden in the MSBB sample set (FDR < 0.01 for Brodmann area (BM) 22, FDR < 0.05 for BM36).

To test whether KLK7 is involved in Aβ degradation, we examined a specific neutralizing antibody against KLK7 (MAB2624) (Bin et al, 2011). Addition of MAB2624 significantly reduced Aβ degradation activity in CCF-STTG1 cells (Fig 3A and B, and Appendix Fig S3A). Moreover, the conditioned medium from COS-1 cells overexpressing KLK7 showed the degradation activity against naturally secreted Aβ (Fig 3C and Appendix Fig S3B), which was abolished by the addition of MAB2624. Finally, coincubation of recombinant purified human KLK7, but not KLK6, derived from either mammalian cells or bacteria with 7PA2-derived Aβ or the synthetic Aβ resulted in a significant Aβ degradation in vitro (Fig 3D and Appendix Fig S3C–E), indicating that KLK7 is directly involved in the Aβ degradation. Consistent with previous result (Shropshire et al, 2014), KLK7 degraded synthetic Aβ fibrils (Fig 3E). To further elucidate the role of KLK7 in murine astrocyte, we analyzed the primary glial cells obtained from rat pups. This culture is mainly comprised of primary astrocytes, but still contained the primary microglia. Notably, Klk7-positive puncta were detected only in Aldh1L1-positive primary astrocytes, but not in Iba-1-positive microglia (Appendix Fig S4A). Conditioned medium from the primary glial cell culture also showed chymotrypsin-type serine protease-dependent Aβ degradation activity inhibited by the neutralizing antibody against KLK7 (Fig 4A and B, and Appendix Fig S4B). Other proteases might be also involved in the Aβ degradation because the MAB2624 showed partial inhibition in this assay. However, these results implicated that Klk7-dependent Aβ degradation activity was associated with the astrocytes.

### Importance of KLK7 in the brain Aβ pathology

We then generated homozygous Klk7 knockout mice (Klk7−/−) using ES cells carrying Klk7tm1(KOMP)Vlcg VelociGene deletion allele. Klk7−/− mice appeared normal and were fertile, and their expression of Klk7 mRNA was completely abolished (Appendix Fig S5A–C). Moreover, MAB2624 failed to inhibit the Aβ degradation in the conditioned medium from the primary glial culture obtained from Klk7−/− mice, indicating that MAB2624 specifically inhibited the proteolytic activity of KLK7 protein in the degradation assay (Appendix Fig S5D). However, no commercial antibody detected the endogenous brain KLK7 protein on the immunoblot. Then, we analyzed the levels of endogenous murine brain Aβ and found 1.4-fold to twofold increase in both male and female Klk7−/− mouse brains (Appendix Fig S5E). No change was observed in the expression levels of APP, proteolytic fragments of APP, ADAM10, BACE1,

## Figure 1. Pharmacological analysis of Aβ degradation activity in the conditioned medium of astrocytoma cell lines.

A    Aβ degradation activity in the conditioned media from CCF-STTG1 and U87 cells. The media were incubated for 24 h with the conditioned medium of 7PA2 cells. Remaining Aβ in the mixture was visualized by immunoblotting. Aβ secreted from 7PA2 cells completely disappeared after 24 h, and this was inhibited by the addition of a complete protease inhibitor cocktail.

B    Aβ remained in the mixed medium by the addition of diisopropyl fluorophosphates or complete protease inhibitor cocktail. DIFP, diisopropyl fluorophosphates; PR, phosphoramidon; PepA, pepstatin A.

C    Effect of tosyl-*L*-lysyl-chloromethane hydrochloride or tosyl phenylalanyl chloromethyl ketone on Aβ degradation activity in the conditioned medium of CCF-STTG1 cells. TLCK, tosyl-*L*-lysyl-chloromethane hydrochloride; TPCK, tosyl phenylalanyl chloromethyl ketone.

D    Effect of matrix metalloprotease inhibitor, GM6001, on Aβ degradation activity in the conditioned medium of CCF-STTG1 cells.

E    Effect of known Aβ degrading serine protease inhibitors on Aβ degradation activity in the conditioned medium from CCF-STTG1 cells. AcMet, acetylmethionine; αplasmin, α2-anti-plasmin.

F    Effect of zinc ions on Aβ degradation activity in the conditioned medium of CCF-STTG1 cells.

Source data are available online for this figure.

γ-secretase, ApoE, neprilysin, insulin-degrading enzyme or matrix metalloprotease-9 (Appendix Fig S5F). Injection of recombinant KLK7 protein into hippocampi of wild-type mouse significantly reduced the murine brain Aβ levels, supporting the notion that Klk7 is physiologically involved in the catabolism of brain Aβ (Appendix Fig S5G).

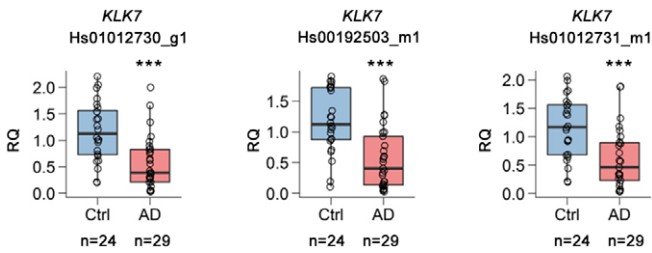

## Figure 2. Expression analyses of *KLK7* mRNA in the brains of AD patients.

Relative mRNA expression levels of *KLK7* in human autopsied brain samples from control (Ctrl) subjects ($n = 24$) and AD patients ($n = 29$) are shown by box-and-whisker plots. The box plots indicate median (solid line in the middle) ± 25th percentile. The whiskers indicate the smallest or highest values that are within 1.5 times the interquartile range below the 25th or above the 75th percentile, respectively. Three different TaqMan probes for *KLK7* mRNA (Hs01012730_g1, Hs00192503_m1, Hs01012731_m1) were used for the qRT–PCR analysis. Relative expression of *KLK7* mRNA was standardized by *GUSB* mRNA levels (TaqMan probe Hs99999908_m1), which exhibited comparable expression between AD patients and control subjects. Statistical analysis was performed by Mann–Whitney *U*-test between control and AD. ***$P < 0.001$.

Then, we examined the amyloid pathology of *App* knockin mice (Saito *et al*, 2014, 2016) under a *Klk7*-null background. Homozygous $App^{NL-G-F/NL-G-F}$ mice showed cortical amyloid deposition at 1–2 months of age, and thioflavin S-positive plaques that were surrounded by glial fibrillary acidic protein (GFAP)-positive astrocytes appeared at 6–7 months of age. As the Arctic mutation in the knockin allele promotes aggregation (Nilsberth *et al*, 2001) and decreases the immunoreactivity in the sandwich enzyme-linked immunosorbent assay (Saito *et al*, 2014), we compared the brain Aβ levels by immunoblot (Fig 5A–C and Appendix Fig S6A–C). We observed a significant increase in the levels of Tris buffer-soluble human Aβ in the brains of $App^{NL-G-F/NL-G-F}$ mice at 3 months of age by genetic *Klk7* ablation without affecting APP and related proteins (Appendix Fig S7A). Moreover, amounts of insoluble Aβ (i.e., SDS-soluble and formic acid-soluble) were also increased. Consistent with these biochemical analyses, brain amyloid deposition was drastically increased (5.6-fold) in the brains of $App^{NL-G-F/NL-G-F}$; $Klk7^{-/-}$ mice (Fig 5D and E). These data implicated a significant impact of loss of *Klk7* not only on the biochemical Aβ economy, but on the deposition pattern of the Aβ plaques.

We further analyzed the pathological changes related to the Aβ deposition. We observed the accelerated phosphorylation of murine endogenous tau (Appendix Fig S8A and B) as well as formation of BACE1-accumulated dystrophic neurites (Appendix Fig S8C) (Kandalepas *et al*, 2013) in the brains of $App^{NL-G-F/NL-G-F}$; $Klk7^{-/-}$ mice. In addition, increased thioflavin S-positive amyloid plaques (Fig 5E) as well as GFAP-positive gliosis around the plaques were

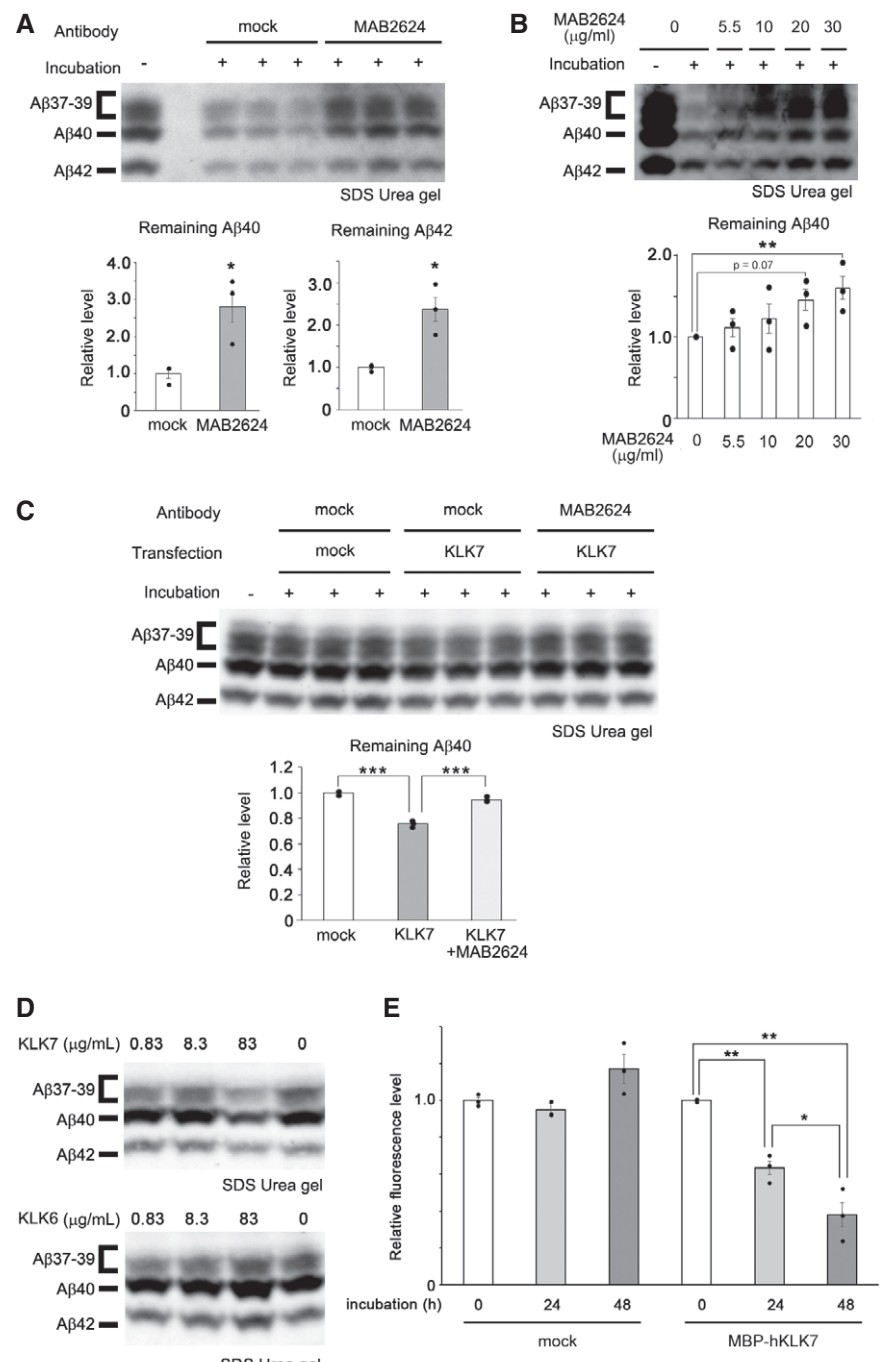

**Figure 3. KLK7 is involved in the Aβ degradation activity.**

A   Inhibition of Aβ degradation activity of CCF-STTG1 cells with MAB2624, a KLK7-neutralizing antibody. Quantification of the relative remaining Aβ40 and Aβ42 in a mixture of normal culture medium and conditioned medium is shown below the blot (*n* = 3, mean ± s.e.m., **P* < 0.05 by Student's *t*-test).

B   Dose-dependent inhibition of Aβ degradation activity of CCF-STTG1 cells with MAB2624. Quantification of relative remaining Aβ40 in the mixture of normal culture medium and conditioned medium is shown below the blot (*n* = 3, mean ± s.e.m., ***P* < 0.01 by Tukey's test).

C   Aβ degradation activity in the conditioned medium of COS-1 cells expressing human KLK7. Quantification of relative remaining Aβ40 in the mixture of normal culture medium and conditioned medium is shown below the blot (*n* = 3, mean ± s.e.m., ****P* < 0.001 by Tukey's test).

D   Aβ degradation activity of the recombinant KLK7 protein. Immunoblot analysis of the remaining Aβ in the mixture of normal cultured medium and conditioned medium of 7PA2 cells, and recombinant human KLK7 and KLK6 protein is shown.

E   *In vitro* degradation of preformed Aβ fibril by purified MBP-tagged hKLK7 protein. Amounts of Aβ fibrils were measured by thioflavin T fluorescence, and relative fluorescence levels at each time point were shown (*n* = 3, mean ± s.e.m., **P* < 0.05, ***P* < 0.01 by Tukey's test).

Source data are available online for this figure.

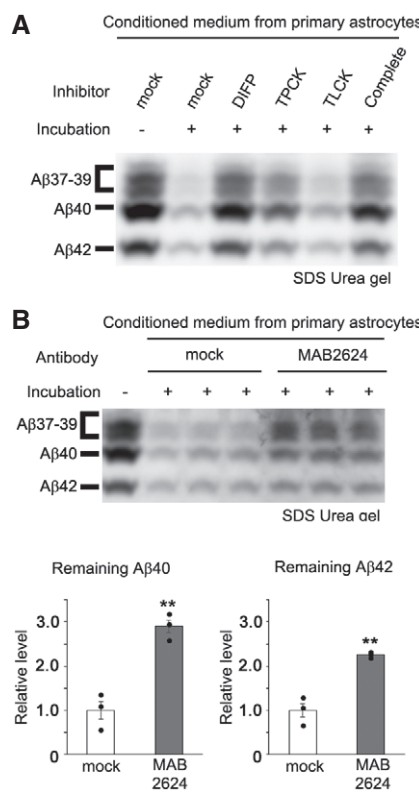

**Figure 4.  Klk7 is involved in the Aβ degradation activity detected in the conditioned medium of primary astrocytes.**

A   Aβ degradation activity in the conditioned medium of primary astrocytes. Note that diisopropyl fluorophosphates and tosyl phenylalanyl chloromethyl ketone inhibited this activity in a similar manner to those observed in the conditioned medium of CCF-STTG1 cells. DIFP, diisopropyl fluorophosphates; TLCK, tosyl-L-lysyl-chloromethane hydrochloride; TPCK, tosyl phenylalanyl chloromethyl ketone.

B   Effect of the KLK7-neutralizing antibody MAB2624 on the Aβ degradation activity of primary astrocytes. Quantification of relative remaining Aβ40 and Aβ42 in the mixture of normal culture medium and conditioned medium is shown below the blot ($n = 3$, mean ± s.e.m., **$P < 0.01$ by Student's *t*-test).

Source data are available online for this figure.

detected in the congenic mice (Appendix Fig S8D), supporting our notion that amyloid-induced neuritic/astrocytic changes were augmented in the congenic mice. Notably, although the expression of GFAP was significantly increased, Aldh1L1 levels were not affected by *Klk7* gene ablation, suggesting that there was an activation without a change in astrocyte number (Appendix Fig S7B). Collectively, these results indicate that Klk7 is a crucial component of brain Aβ economy and attenuates brain amyloid pathology in AD model mice.

### Activation of Klk7 expression by Aβ treatment and memantine

KLK7 is a terminal protease in the kallikrein-related peptidase cascade and is directly activated by KLK5-mediated prodomain removal (Sotiropoulou *et al*, 2009). Importantly, the upregulation of KLK5 and KLK7 activity by loss of the endogenous KLK inhibitor serine peptidase inhibitor Kazal type 5 (SPINK5) causes atopic

dermatitis-associated diseases, including Netherton syndrome (Furio & Hovnanian, 2014). However, *Klk7* mRNA expression is selectively regulated by cytokines in the skin (Morizane *et al*, 2012), suggesting that transcription of *Klk7* mRNA is controlled by a specific mechanism. In fact, *Klk7* mRNA expression was significantly and selectively increased in $App^{NL-G-F/NL-G-F}$ mice (Fig 6A). Moreover, mRNA expression of *Klk7*, but not other Aβ-degrading enzymes (i.e., matrix metalloproteases, neprilysin, and insulin-degrading enzyme), was further specifically augmented in an age-dependent manner (Fig 6B and Appendix Fig S9). Intriguingly, treatment of primary astrocytes with Aβ42, but not with lipopolysaccharide, significantly increased the expression of *Klk7* (Fig 6C and D). Although we are unable to exclude the possibility that lipopolysaccharide has some regulatory role in the *Klk7*, these results suggest that *Klk7* transcription was selectively modulated by Aβ in the astrocytes and that selective augmentation of *Klk7* expression is possible.

We recently investigated the effect of the memantine, which is the Food and Drug Administration-approved anti-dementia drug and an *N*-methyl-D-aspartate receptor antagonist, on the Aβ metabolism in the primary cultures (Ito *et al*, 2017). We noticed that the memantine treatment significantly reduced the spiked human Aβ only in the conditioned medium of primary neuron and glia coculture, but not in that of primary neuronal culture (Fig 7A and B). In addition, we as well as others have found that the chronic treatment of Tg2576 *APP* transgenic mice with memantine reduced Aβ levels (Dong *et al*, 2008; Ito *et al*, 2017). Thus, we hypothesized that memantine would affect *Klk7* mRNA expression in the astrocytes and Aβ deposition in the brain. Supporting this notion, *Klk7* mRNA level was specifically upregulated in the brains of memantine-treated Tg2576 mice (Fig 7C). Moreover, memantine increased the *Klk7* mRNA in the primary astrocytes. However, the *N*-methyl-D-aspartate receptor antagonist MK-801 failed to induce *Klk7* mRNA in the primary astrocytes (Fig 7D). Intriguingly, glutamate as well as *N*-methyl-D-aspartate treatment significantly reduced the *Klk7* mRNA level, indicating the possibility that MK-801-insensitive, astrocytic glutamate signaling is involved in the regulation of *Klk7* expression (Fig 7E). We also found that Aβ-degrading efficiency was significantly enhanced by conditioned medium obtained from the memantine-treated primary astrocytes (Appendix Fig S10). This effect was abolished in the astrocytes from *Klk7*-knockout mice (Fig 7F). These data indicate that astrocytic *Klk7* expression is regulated by Aβ in the murine brain and is selectively upregulated by memantine.

## Discussion

In this study, we identified that KLK7 is a crucial astrocytic key component in the regulation of brain Aβ economy. We observed 1.4-fold to twofold increase of endogenous Aβ levels in the brains of $Klk7^{-/-}$ mice, which is almost comparable to the effects by genetic deletion of neprilysin or insulin-degrading enzyme *in vivo* (Saido & Leissring, 2012). Moreover, *Klk7* deletion increased the thioflavin S-positive amyloid deposition in AD model mice. Thus, KLK7 is one of critical Aβ-degrading enzymes, and this indicates that astrocytes are involved in regulation of Aβ pathogenesis *via* Klk7 pathway. It was reported that KLK7 is capable to cleave the hydrophobic core motif of Aβ fibrils and reduces the cell toxicity *in vitro* (Shropshire *et al*,

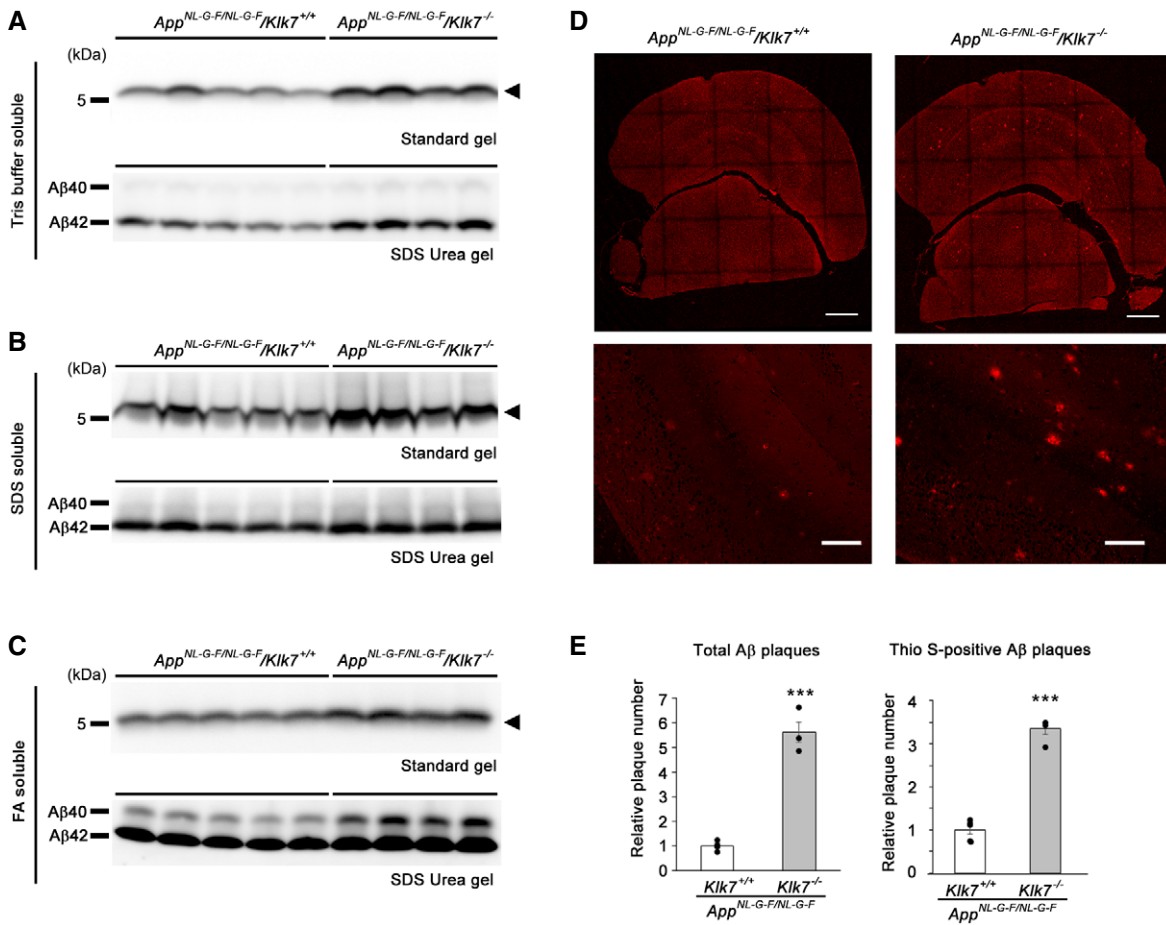

**Figure 5. *Klk7* gene deficiency increases the brain Aβ levels and amyloid pathology in App<sup>NL-G-F/NL-G-F</sup> mice.**

A–C    Biochemical analyses of human Aβ in Tris buffer-soluble (A), SDS-soluble (B), and formic acid-soluble (C) fractions of the brains of 3-month-old male *App*<sup>NL-G-F/NL-G-F</sup>;
         *Klk7*<sup>−/−</sup> mice. Arrowheads indicate total Aβ. Quantification of relative levels of Aβ is shown in Appendix Fig S6.
D        Immunohistochemical analysis of the brains of 3-month-old male *App*<sup>NL-G-F/NL-G-F</sup>; *Klk7*<sup>−/−</sup> mice using the 82E1 antibody (red). Magnified images were shown below.
         Scale bar, 100 μm.
E        Quantification results of cortical 82E1-positive total Aβ plaques (left) and the cortical thioflavin S-positive Aβ plaques (right) are shown (*n* = 3 or 4, mean ± s.e.m.,
         ****P* < 0.001 by Student's *t*-test).

Source data are available online for this figure.

2014). Consistent with this result, we found that *App*<sup>NL-G-F/NL-G-F</sup>; *Klk7*<sup>−/−</sup> mice showed the accelerated neuritic changes and inflammatory responses (i.e., increased tau phosphorylation, dystrophic neurites, and reactive astrocytes). However, we did not observe notable change in the number of neurons in the brains and the behavior of congenic mice, probably due to the young age of the model mice. Nevertheless, it remains unclear whether upregulation of KLK7 in the brain causes noxious effect by degrading the other substrates. It is important to understand the physiological function and substrate of KLK7 in the brain by a proteomic approach (Yu *et al*, 2015).

Aβ treatment selectively induced the expression of *Klk7 in vitro*, and the *Klk7* expression was correlated with Aβ deposition in AD model mice. Molecular mechanism of upregulation of *KLK7* mRNA by Aβ remains unclear. However, KLK7 has been implicated in the skin inflammation (Furio & Hovnanian, 2014; Prassas *et al*, 2015). Intriguingly, anti-inflammatory Th2 cytokines, such as interleukin-4

and interleukin-13, increase *KLK7* mRNA expression in human keratinocytes (Morizane *et al*, 2012). These data suggest that *Klk7* expression is homeostatically regulated by Aβ-induced inflammatory response in the astrocytes. Notably, expression of the other Aβ-degrading enzymes, matrix metalloprotease-2, and matrix metalloprotease-9 was also increased in the reactive astrocytes (Yin *et al*, 2006), although the level of insoluble Aβ was not affected in the brains of *Mmp2* or *Mmp9* knockout mice. This also suggests that KLK7 plays a primary role in the clearance of aggregated, oligomeric form of Aβ *in vivo*. Nevertheless, modulation of astrocytic response in the astrocytes to increase KLK7 expression would be a novel therapeutic approach to reduce Aβ deposition (De Strooper & Karran, 2016).

However, in contrast to the model mice, we found that *KLK7* mRNA was significantly decreased in AD patients with significant Aβ deposition in the brain. Importantly, the AD model mouse did not reflect all hallmarks of human AD brain (e.g., formation of

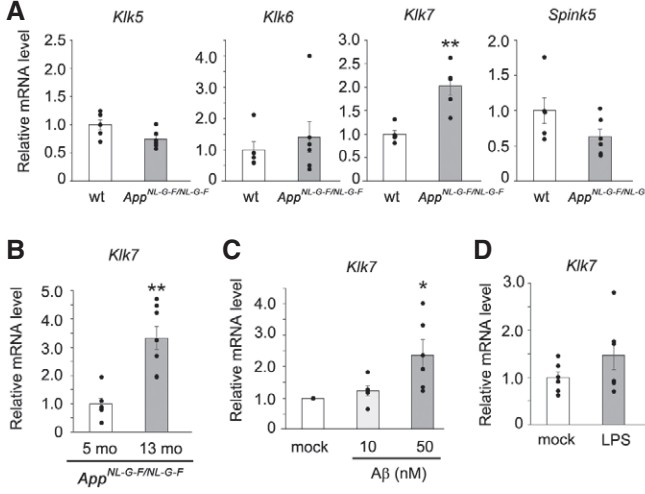

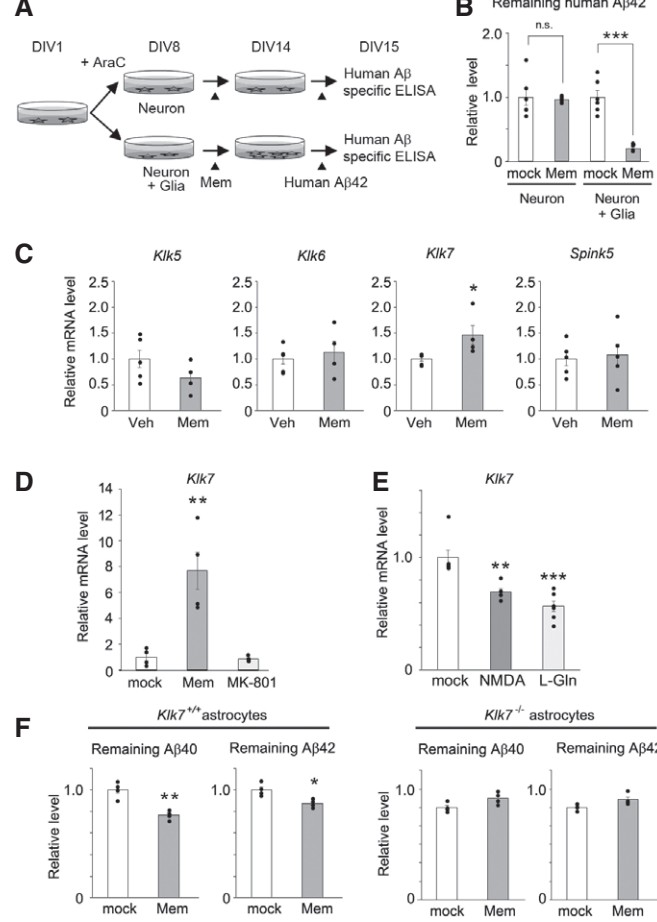

**Figure 6.  Upregulation of *Klk7* expression by Aβ in the astrocytes.**

A  Relative levels of *Klk5*, *Klk6*, *Klk7*, and *Spink5* mRNA in the brains of 5-month-old male *App*<sup>NL-G-F/NL-G-F</sup> mice (*n* = 5 or 6, mean ± s.e.m., \*\**P* < 0.01 by Student's *t*-test).

B  Levels of *Klk7* mRNA in 5-month-old and 13-month-old male *App*<sup>NL-G-F/NL-G-F</sup> mice (*n* = 5 or 6, mean ± s.e.m., \*\**P* < 0.01 by Student's *t*-test).

C  Effect of the Aβ peptide on *Klk7* mRNA expression in primary astrocytes obtained from wild-type mice (*n* = 6, mean ± s.e.m., \**P* < 0.05 by Tukey's *t*-test).

D  Effect of 1 μg/ml lipopolysaccharide on *Klk7* mRNA expression in primary astrocytes obtained from wild-type mice (*n* = 6, mean ± s.e.m., \**P* < 0.05 by Student's *t*-test).

tangles and major neuronal loss). Thus, these pathological changes might affect the expression of *KLK7* mRNA. Moreover, longitudinal studies revealed that Aβ deposition starts 10–15 years before onset of clinical symptoms (De Strooper & Karran, 2016). Thus, other possibility is that the prolonged inflammatory response by Aβ deposition might alter the phenotypes of astrocytes to reduce the *KLK7* expression in human brains, although the mechanism underlying this pathological phenotype remains unclear. Notably, expression of KLK7 mRNA was regulated by methylation of histones in cancer cell lines (Raju *et al*, 2016). It would be important to analyze epigenetic regulation of *KLK7* and/or activation status of the astrocytes in the AD brains.

Most important finding of this study is that *Klk7* expression was increased by memantine treatment *in vitro* and *in vivo*. Memantine would have a great potential in combination therapy to facilitate amyloid clearance, for instance in addition to immunotherapy or even as monotherapy after successful removal of amyloid by immunotherapy. Obtaining mechanistic insight into the selective *Klk7* induction by memantine would pave the road toward developing astrocyte-targeted AD therapeutics. Notably, authentic *N*-methyl-D-aspartate receptor antagonist MK-801 failed to affect the *Klk7* mRNA expression, whereas glutamate as well as *N*-methyl-D-aspartate decreased the *Klk7* level. This result suggested the possibility that the *N*-methyl-D-aspartate receptor in astrocyte shows a different response for these antagonists to that in neurons. In fact, unusual subunit composition of glial *N*-methyl-D-aspartate receptors distinct from that in neurons has been highlighted (Dzamba *et al*, 2013), raising the possibility that MK-801-insensitive novel

**Figure 7.  Effect of memantine on Klk7-dependent Aβ degradation activity.**

A, B  Effect of memantine in human Aβ spike experiment. Primary neuron culture or neuron and glia coculture were treated with or without 30 μM of memantine (Mem) at DIV8. 30 nM of synthetic human Aβ42 peptide was spiked to these cultures. Levels of remaining human Aβ42 at days *in vitro* measured by human Aβ-specific sandwich enzyme immunosorbent assay are shown in (B) (*n* = 6, mean ± s.e.m., \*\*\**P* < 0.001 by Student's *t*-test).

C  Relative levels of *Klk5*, *Klk6*, *Klk7*, and *Spink5* mRNA in the brains of memantine (Mem)-treated female Tg2576 mice (*n* = 5 or 4, mean ± s.e.m., \**P* < 0.05 by Student's *t*-test). Veh, vehicle.

D  Effect of 30 μM memantine or 10 μM MK-801 on *Klk7* mRNA expression in primary astrocytes obtained from wild-type mice (*n* = 5 or 6, mean ± s.e.m., \*\**P* < 0.01 by Tukey's test).

E  Effect of 20 μM L-glutamate or 50 μM *N*-methyl-D-aspartate on *Klk7* mRNA expression in primary astrocytes obtained from wild-type mice (*n* = 5 or 6, mean ± s.e.m., \*\**P* < 0.01 and \*\*\**P* < 0.001 by Tukey's test).

F  Effect of 30 μM memantine on Aβ degradation activity in the conditioned medium of primary astrocytes obtained from wild-type or *Klk7*<sup>−/−</sup> mice. Remaining Aβ40 and Aβ42 are quantified (*n* = 4, mean ± s.e.m., \**P* < 0.05, \*\**P* < 0.01 by Student's *t*-test).

astrocytic *N*-methyl-D-aspartate receptor is involved in the regulation of *Klk7* expression. Intriguingly, memantine induced the expression of Th2 cytokines in T cells irrespective of traditional *N*-methyl-D-aspartate receptor expression (Kahlfuss *et al*, 2014). Thus, memantine might modulate anti-inflammatory cytokine signaling in

astrocytes to increase *Klk7* expression *via* novel mechanism. Importantly, significant adverse effects of memantine on the skin have not been reported, suggesting the possibility that memantine treatment was not sufficient for the skin disease phenotype by overactivation of KLK7 (Hansson *et al*, 2002). Nevertheless, understanding the regulatory mechanism of KLK7-mediated Aβ degradation at the molecular level should provide novel insights into the pathological role of astrocytes, as well as their possibility as a therapeutic target against AD.

# Materials and Methods

### Analysis of brains of AD patients

The patient demographics are shown in Appendix Fig S2A. Total RNA from brain tissues was extracted with a TRIzol Plus RNA Purification System (Life Technologies) from frozen brain tissues of the frontal cortex. The RNA integrity number (RIN) was determined by a 2100 Bioanalyzer (Agilent Technologies). Samples were subjected to RT–qPCR amplification with a TaqMan Gene Expression Assay (Life Technologies) on an ABI PRISM 7900 HT instrument (Applied Biosystems, Carlsbad, CA, USA) as previously described (Miyashita *et al*, 2014). Relative gene expression levels were calculated as the cycle difference by means of the delta-delta Ct method using three internal controls genes including *GUSB*, *RPS17*, and *CASC3* (Appendix Fig S2B) (Miyashita *et al*, 2014). The subjects were neuropathologically grouped according to the neurofibrillary tangle staging of Braak and Braak (Braak & Braak, 1991).

### Data availability

We analyzed two public RNAseq datasets deposited at AMP-AD knowledge portal (https://www.synapse.org/#!Synapse:syn2580853): the Mayo sample set (Allen *et al*, 2016) and MSBB studies. The Mayo study comprises 156 temporal cortex samples from 80 patients with AD and 76 controls. We assessed *KLK7* expression of the temporal cortex between AD patients and controls by a simple model adjusting for key covariates: age at death, gender, RIN, source, and flow cell. For the MSBB study, we selected 228 samples from BM22 and 206 samples from BM36 excluding the samples without the information of Braak NFT stage and RIN. We downloaded raw read-count data and applied them to the R package, *DESeq2* (version 1.14.1) to explore alteration of gene expression associated with the amyloid plaque burden. More specifically, we divided degrees of neuritic plaque density into five categories at equal intervals and compared gene expression levels among the five categories adjusting for age at death, gender, race, and RIN.

### Animals

*App*^*NL-G-F/NL-G-F*^ mice (Saito *et al*, 2014) were originally generated at RIKEN Brain Science Institute, Japan. The sperm of *Klk7*^*tm1(KOMP)Vlcg/+*^ heterozygous mice (project ID: VG14816) used for this study was generated by the trans-NIH Knockout Mouse Project [VelociGene at Regeneron Inc. (U01HG004085) and the CSD Consortium (U01HG004080)] and obtained from the KOMP Repository at UC Davis and CHORI (U42RR024244) (www.komp.org). *In vitro*

fertilization using the sperm was achieved at the Center for Disease Biology and Integrated Medicine, Graduate School of Medicine, The University of Tokyo. Maintenance of Tg2576 mice (Hsiao *et al*, 1996) and chronic administration of memantine (20 mg/kg/day) to 8-month-old female Tg2576 mice for 1 month by p.o. were performed at Daiichi-Sankyo Co. Ltd. Genomic DNAs of mice were obtained from tails by alkaline lysis. Genotypes were determined by genomic PCR using following primers: 5′-gcagcctctgttccacatacacttca-3′ (Reg-NeoF) and 5′-accacacaacaacagtctctcttgc-3′ (Reg-Klk7-R) for *Klk7* knockout locus; 5′-ctgaggcaatctcaccgtctgg-3′ (Reg-Klk7-wtR) and 5′-cagagtgcccagaagatcaagg-3′ (Reg-Klk7-wtF) for *Klk7* wild-type locus; 5′-atctcggaagtgaagatg-3′ (E16WT) and 5′-tgtagatgagaacttaac-3′ (WT) for *App* wild-type locus; and 5′-atctcggaagtgaatcta-3′ (E16MT) and 5′-cgtataatgtatgctatacgaag-3′ (loxP) for *App* knockin locus.

### Antibodies and chemicals

The following primary antibodies were used in this study: anti-human Aβ 82E1 (1:2,500, IBL #10323), anti-human Aβ 6E10 (1:2,000, BioLegend #SIG-39300), anti-KLK7 (1:2,000, Abcam ab28309), anti-KLK7 antibody for neutralization (20 μg/ml, R&D Systems MAB2624), anti-KLK7 for immunostaining (1:40, R&D Systems AF2624), anti-neprilysin/CD10 (1:2,000, R&D Systems AF1126), anti-IDE (1:2,000, Merck Millipore ST1120), anti-GFAP (1:1,000, Sigma-Aldrich G3893), anti-Aldh1L1 for immunostaining (1:5, NeuroMab 75-164), anti-Aldh1L1 for immunoblotting (1:5, NeuroMab 75-140), anti-BACE1 BACE1c (1:1,000, IBL #18711), anti-APP APPc (1:1,000, IBL 18961), anti-nicastrin N1660 (1:1,000, Sigma-Aldrich N1660), anti-ADAM10 (a kind gift from Dr. Paul Saftig) (Suzuki *et al*, 2012), anti-calnexin C-terminus (1:1,000, Enzo Life Sciences ADI-SPA-860), anti-total Tau Tau-5 antibody (1:1,000, BioLegend # SIG-39413), anti-Phospho-PHF-tau pThr231 AT180 antibody (1:1,000, Invitrogen MN1040), Anti-Phospho-PHF-tau pThr181 AT270 antibody (1:1,000, Invitrogen MN1050), anti-MMP-9 TP221 (1:1,000, Torrey Pines Biolabs), and anti-α-tubulin (1:5,000, Sigma-Aldrich DM1A). For isotype control of MAB2624 (mouse IgG2a), we used same concentration (final 30 μg/ml) anti-V5 Tag (Thermo Fisher #R960-25) and anti-LR11 (BD Biosciences #611860) monoclonal antibodies.

The following chemicals were used in this study: thioflavin S (Sigma-Aldrich, stock solution was dissolved in 50% ethanol, final concentration was 0.05%), diisopropyl fluorophosphates (WAKO, stock solution was dissolved in 2-propanol, final concentration was 0.5 mM), E-64 (Roche Applied Science, stock solution was dissolved in 50% ethanol, final concentration was 10 μg/ml), ethylenediaminetetraacetic acid (DOJINDO, stock solution was dissolved in distilled water, final concentration was 1 mM), phosphoramidon (Nacalai Tesque, stock solution was dissolved in distilled water, final concentration was 10 μg/ml), pepstatin A (Sigma-Aldrich, stock solution was dissolved in 10% acetic acid, final concentration was 1 μg/ml), complete protease inhibitor cocktail (Roche Applied Science), PhosSTOP phosphatase inhibitor cocktail (Roche Applied Science), α2-anti-plasmin (WAKO, stock solution was dissolved in distilled water, final concentration was 70 μg/ml), acetylmethionine (Tokyo Chemical Industry, stock solution was dissolved in distilled water, final concentration was 1 mM), tosyl-*L*-lysyl-chloromethane hydrochloride (WAKO, stock solution was dissolved in ethanol, final concentration was

100 μg/ml), tosyl phenylalanyl chloromethyl ketone (WAKO, stock solution was dissolved in DMSO, final concentration was 100 μg/ml), GM6001 (Enzo Life Science, stock solution was dissolved in DMSO, final concentration was 25 μM), MK-801 (Sigma-Aldrich, stock solution was dissolved in distilled water, final concentration was 10 μM), *N*-methyl-D-aspartate (Sigma-Aldrich, stock solution was dissolved in distilled water, final concentration was 50 μM), L-glutamine (KANTO, stock solution was dissolved in distilled water, final concentration was 20 μM), lipopolysaccharide from *Escherichia coli* (Imgenex, stock solution was dissolved in distilled water, final concentration was 1 μg/ml), and DRAQ5™ far-red fluorescent DNA dye (ImmunoChemistry Technologies, final concentration was 1:1,000 dilution). Memantine (stock solution was dissolved in distilled water, final concentration was 30 μM) was provided by Daiichi-Sankyo Co. Ltd.

### Cell culture and transfection

CCF-STTG1, U-87, and BE(2)-C cells were purchased from European Collection of Authenticated Cell Cultures. MG-6 cell (Takenouchi *et al*, 2005; Nakamichi *et al*, 2006) was provided by the RIKEN BRC through the National Bio-Resource Project of the MEXT, Japan. COS-1 (Maruyama *et al*, 1990), HEK293 (Morohashi *et al*, 2002), Neuro2a (Tomita *et al*, 1997), H4 (Asai *et al*, 2010), and SH-SY5Y (Takasugi *et al*, 2003) cells were kindly provided from Dr. Kei Maruyama (Saitama Medical University). BV-2 (Blasi *et al*, 1990) and 7PA2 cells (Podlisny *et al*, 1995) were gifts from Drs. Makoto Michikawa (Nagoya City University) and Edward Koo (University of California, San Diego), respectively. Cells were cultured in Dulbecco's modified Eagle's medium with high glucose (WAKO) supplemented with 10% heat-inactivated fetal bovine serum (FBS; Hyclone), 50 units/ml penicillin (Invitrogen), and 50 mg/ml streptomycin (Invitrogen) at 37°C under humidified air containing 5% $CO_2$. alamarBlue Cell viability assay (Thermo Fisher Scientific) was performed as manufacturer's instruction. We routinely check mycoplasma contamination by DAPI staining and PCR analysis. Primary dissociated cortical glial cultures were prepared from mice at postnatal day (P) 1–3 or P18 Wistar rats and maintained in Dulbecco's modified Eagle's medium containing 10% FBS, as previously described (Fukumoto *et al*, 1999; Suzuki *et al*, 2012). To separate primary astrocyte and microglia, the glial culture was gently shaken for 24 h at 37°C under humidified air containing 5% $CO_2$. Primary microglia detached to the medium were then replated to different plate. Human *KLK7* and *KLK6* cDNAs were amplified from DNAFORM clone 100008376 (GenBank accession no. DQ893916) and Human brain Total RNA (TaKaRa, #636539), respectively, using KOD-Plus-Neo DNA polymerase (TOYOBO). For the overexpression experiment in mammalian cell, cDNA was inserted into the pEF6/V5-His-TOPO vector (Invitrogen). Transfection into COS-1 cells using FuGene 6 (Roche Applied Science) was described previously (Suzuki *et al*, 2012). For purification of MBP-tagged proteins, cDNA was inserted into pMAL-p2x vector (New England Biolabs) (Miyamoto *et al*, 2011). Plasmids are transformed into *E. coli* Rosetta 2(DE3) (Novagen), and periplasmic expression was induced by 0.1 mM IPTG. Overexpressed proteins were purified by amylose column with maltose according to manufacturer's instructions.

### Aβ degradation assay

7PA2 cells were cultured at confluency for 24 h, and their conditioned medium was collected. After centrifugation at 5,900 × *g* for 10 min at 4°C, the supernatant was stored at −80°C until use. Aβ levels were determined by ELISA or immunoblotting, using the urea/SDS–PAGE gel system, as described previously (Tomita *et al*, 1997; Ohki *et al*, 2011). Synthetic Aβ peptides were obtained from Peptide Institute (Aβ40 and Aβ42) and AnaSpec (Aβ37, Aβ38, and Aβ39). For the Aβ degradation assay, the 7PA2 medium was coincubated with the same amount of conditioned medium from CCF-STTG1, U-87, and primary glial cells at 37°C for 24 h. Recombinant human KLK6 and KLK7 proteins purchased from R&D Systems were diluted with Hank's balanced salt solution (HBSS). 7 μl of 5 nM synthetic Aβ40 peptide was mixed with 49 μl of HBSS containing KLK proteins at the indicated concentrations and incubated at 37°C for 24 h.

For human Aβ spike experiment, a rat primary neuron culture was obtained by the treatment of 2 μM of AraC to the primary cells isolated from rat brain on days *in vitro* 1 as previously described (Fukumoto *et al*, 1999; Suzuki *et al*, 2012). The primary culture without AraC treatment was defined as the neuron + glia culture. At days *in vitro* 8, 30 μM of memantine or water was added to the cultures. At days *in vitro* 14, 30 nM of synthetic human Aβ42 peptide was added to the primary neuron or neuron/glial culture. At days *in vitro* 15, levels of remaining human Aβ42 in the conditioned medium were measured using a human Aβ-specific sandwich ELISA kit (#296-64401, WAKO) (*n* = 6, mean ± s.e.m.).

### Injection of recombinant proteins into the hippocampal region of mouse brains

MBP and MBP-hKLK7 proteins were injected into the hippocampi (Hori *et al*, 2015) (anterior–posterior −2.5 mm, medial–lateral ±2.0 mm, dorsal–ventral −1.8 mm from Bregma) in 5-month-old wild-type mice. Three hours after injection, the hippocampi were extracted and homogenized in Tris buffer. Aβ40 level in Tris-soluble fraction was measured by two-site ELISA system, Human/Rat β-Amyloid (40) ELISA Kit (#294-62501, WAKO).

### Immunocytochemistry and immunohistochemistry

Cells were fixed with 4% paraformaldehyde in phosphate-buffered saline (8 mM $Na_2HPO_4$, 2 mM $NaH_2PO_4$, and 131 mM NaCl) for 20 min, incubated in phosphate-buffered saline containing 0.1% Triton X-100 for 15 min, and stained for 2 h with primary antibodies at room temperature. The cells were then washed and incubated for 1 h with Alexa Fluor-conjugated secondary antibodies (Invitrogen), 4′,6-diamidino-2-phenylindole (DAPI), and/or DRAQ5. For immunohistochemistry, 3-month-old mice were anesthetized and perfused with 10 ml PBS, sacrificed by decapitation, and their brains fixed by soaking them in PBS containing 4% paraformaldehyde overnight. The brains were then rinsed in PBS and then serially dehydrated in 70, 80, 90, and 99% ethanol (WAKO), and then, the ethanol was replaced by incubation in xylene (WAKO) twice and then further replaced by incubation in paraffin (WAKO) three times. Paraffin sections (4 μm) were prepared using a microtome (Microedge Instruments, Inc.). Paraffin sections were soaked in xylene three times for 5 min, in ethanol (99, 90, 80, and 70%) for 1 min

each, and then boiled with 0.1 M citrate buffer solution. The sections were blocked with PBS containing 10% cow serum for 30 min and then incubated with primary antibodies overnight, followed by incubation with Alexa 488 or 546-conjugated secondary antibodies (Invitrogen) for 2 h. For thioflavin S staining, sections were stained with 0.05% thioflavin S in 50% ethanol solution for 10 min and washed with 70% ethanol five times. The samples were mounted with PermaFluor™ Aqueous Mounting Medium (Thermo Scientific) and viewed using a confocal laser-scanning microscope (TCS-SP5, Leica). For the quantification of amyloid plaques, the background was subtracted using ImageJ software and the number of amyloid plaques in the cerebral cortex and hippocampus was obtained by particle analysis. Values were then normalized by area size.

### Preparation of samples for biochemical analyses

For total cell lysates, cells were lyzed in 2% SDS, briefly sonicated, and then incubated at 37°C for 30 min with 1% 2-mercaptoethanol. Mouse brains were homogenized in 10× volumes of Tris buffer (50 mM Tris–HCl pH7.6, 150 mM NaCl, complete protease inhibitor cocktail, and PhosSTOP phosphatase inhibitor cocktail) with 25 strokes using a mechanical homogenizer and centrifuged at $200,000 \times g$ for 20 min at 4°C. The resultant supernatant was collected as the brain Tris buffer-soluble fraction. After addition of the same amount of 2% Triton X-100/Tris buffer, the pellet was homogenized on ice and centrifuged at $200,000 \times g$ for 20 min at 4°C. The resultant supernatant was collected as the brain Triton X-fraction. Then, the same amount of 2% SDS containing Tris buffer was added to the pellet. After homogenization at room temperature, the pellet was incubated for 2 h at 37°C and centrifuged at $200,000 \times g$ for 20 min at 20°C. The resultant supernatant was collected as the brain SDS fraction. Finally, the pellet was sonicated (BRANSON, output: 2, duty cycle: 90, 15 s) with 500 µl of 70% formic acid (WAKO) solution. Samples were centrifuged at $200,000 \times g$ for 20 min at 4°C, and the resultant supernatant was freeze-dried for 2 h (Thermo Scientific, Savant RVT5105). The pellet was dissolved in the same volume of DMSO (WAKO) as the brain weight and stored at −80°C until use. Protein concentrations of the fractions were measured by the BCA protein assay (Pierce).

### Immunological analysis and quantitation of immunoblot band densities

Aβ levels were measured by ELISA using the Human/Rat β-Amyloid (40) ELISA Kit (#294-62501, WAKO) and the Human/Rat β-Amyloid (42) ELISA Kit, High Sensitivity (#292-64501, WAKO) or immunoblotting. For immunoblotting, samples were dissolved in Laemmli sample buffer [final concentration of 1 M Tris–HCl pH 6.8, 20% SDS, 30% glycerol, 1% Brilliant Green (WAKO), 1% CBB-G250 (Nacalai Tesque)] and separated by SDS–PAGE. Gels were transferred to a polyvinylidene difluoride (PVDF) membrane (Millipore). The membranes were incubated in 5% skim milk or PVDF Blocking Reagent for *Can Get Signal*® (TOYOBO), treated with primary antibodies, and then probed with horseradish peroxidase (HRP)-conjugated secondary antibodies (GE Healthcare and Jackson ImmunoResearch), and chemiluminescent signals were acquired using ImageQuant LAS 4000 (GE Healthcare). Band intensities were measured using ImageJ software (NIH). The ratio of each of the proteins to α-tubulin or calnexin was acquired and normalized to the value of the control.

### Quantitative real-time PCR analysis for cultured cell and model mouse

Total mRNAs were isolated and purified from cell cultures using ISOGEN reagent (Nippon Gene). 1 µg of purified mRNAs from each sample was reverse-transcribed into cDNA using ReverTra Ace qPCR RT kit (TOYOBO). cDNA templates (1 µl) were used for the real-time PCR reaction and relative quantification using LightCycler 450 (Roche) using the SYBR Green I protocol. The amount of target mRNA was normalized to *Gapdh* mRNA in each sample. The following primer pairs were used to detect the indicated target genes: 5′-atgggcaatggctaccctg-3′ (forward) and 5′-gttcggttccagaggggtt-3′ (reverse) for *Klk5*; 5′-tgtgcttggttcttgctaaatca-3′ (forward) and 5′-agt gacctgaggtgtagaggg-3′ (reverse) for *Klk6*; 5′-tgggtgcgagccttcttac-3′ (forward) and 5′-gctgtctttagccctggaaacc-3′ (reverse) for *Klk7*; 5′-atg ggaaaacataccgcagtag-3′ (forward) and 5′-gcacatatccgtctctaggtgg-3′ (reverse) for *Spink5*; 5′-ggggcttctgcctctttac-3′ (forward) and 5′-tggtc cttgtttcctgggta-3′ (reverse) for *Mme*; 5′-tcggagccttgcttcaatac-3′ (forward) and 5′-tggggtggtttttctgactg-3′ (reverse) for *Ide*; 5′-ctggacagcc agacactaaag-3′ (forward) and 5′-ctcgcggcaagtcttcagag-3′ (reverse) for *Mmp-9;* and 5′-aggtcggtgtgtgaacggatttg-3′ (forward) and 5′-tgtagaccat gtagttgaggtca-3′ (reverse) for *Gapdh*.

### Statistical analysis

For analyses of human samples, statistical analyses were performed by Mann–Whitney *U*-test as previously described (Miyashita *et al*, 2014). All samples were analyzed in blind and randomized manner. For quantitative immunoblot analysis, immunofluorescence, and qRT–PCR in cells and mice, Student's *t*-test was used for comparisons between two-group data, and Tukey's test was used for multiple group comparisons. Statistical analyses were performed by KyPlot or Excel software. In figures, statistical significance is indicated by $*P < 0.05$, $**P < 0.01$, and $***P < 0.001$. All *P*-values are shown in Appendix Table S1.

### Ethical approval

All procedures performed in studies involving human participants were in accordance with the ethical standards of the institutional and/or national research committee and with the 1964 Helsinki declaration and its later amendments or comparable ethical standards. This study was approved by the Institutional Review Board of Niigata University. All animal experimental procedures were performed in accordance with the guidelines for animal experiments of The University of Tokyo or Daiichi-Sankyo Co. Ltd., and were approved by the Institutional Animal Care and Use Committee/ethics committee of the Graduate School of Pharmaceutical Sciences, The University of Tokyo (protocol no. P25-6) or Daiichi-Sankyo Co. Ltd., Tokyo, Japan.

### Informed consent

Informed consent was obtained from all subjects. The written informed consent was received from participants prior to inclusion in the study. Participants were identified by number, not by name.

**The paper explained**

**Problem**

Several lines of evidence suggest that the decreased clearance of Aβ from brain is related to the pathogenesis of Alzheimer's disease (AD). However, the precise role of astrocytes, which is most abundant cell type in the brain, in Aβ clearance remains unclear.

**Results**

Here, we identified kallikrein-related peptidase 7 (KLK7) as an astrocyte-derived Aβ-degrading enzyme that cleaves not only monomer but aggregated forms of Aβ. *KLK7* mRNA level was significantly reduced in AD brain, and genetic ablation of KLK7 in AD model mice exacerbated the amyloid pathology. We also found that anti-dementia drug memantine can increase its expression and Aβ degradation activity of astrocytes. Thus, memantine would have a great potential to facilitate the astrocyte-mediated proteolytic clearance of the brain amyloid.

**Impact**

Our study would provide novel insights into the pathological role of astrocytes in AD, as well as their possibility as a cellular target in the development of anti-Aβ therapeutics.

**Expanded View** for this article is available online.

## Acknowledgements

The authors are grateful to Drs. Kei Maruyama (Saitama Medical University), Paul Saftig (the Christian-Albrechts-Universität Kiel), Umeharu Ohto (The University of Tokyo), Edward Koo (University of California, San Diego) for valuable reagents, and our current and previous laboratory members for helpful discussions. MayoRNAseq Study data were provided by the following sources: The Mayo Clinic Alzheimer's Disease Genetic Studies, led by Dr. Nilufer Taner and Dr. Steven G. Younkin, Mayo Clinic, Jacksonville, FL using samples from the Mayo Clinic Study of Aging, the Mayo Clinic Alzheimer's Disease Research Center, and the Mayo Clinic Brain Bank. Data collection was supported through funding by NIA grants P50 AG016574, R01 AG032990, U01 AG046139, R01 AG018023, U01 AG006576, U01 AG006786, R01 AG025711, R01 AG017216, R01 AG003949, NINDS grant R01 NS080820, CurePSP Foundation, and support from Mayo Foundation. Study data include samples collected through the Sun Health Research Institute Brain and Body Donation Program of Sun City, Arizona. The Brain and Body Donation Program is supported by the National Institute of Neurological Disorders and Stroke (U24 NS072026 National Brain and Tissue Resource for Parkinson's Disease and Related Disorders), the National Institute on Aging (P30 AG19610 Arizona Alzheimer's Disease Core Center), the Arizona Department of Health Services (contract 211002, Arizona Alzheimer's Research Center), the Arizona Biomedical Research Commission (contracts 4001, 0011, 05-901, and 1001 to the Arizona Parkinson's Disease Consortium), and the Michael J. Fox Foundation for Parkinson's Research. MSBB data were generated from postmortem brain tissue collected through the Mount Sinai VA Medical Center Brain Bank and were provided by Dr. Eric Schadt from Mount Sinai School of Medicine. This work was supported in part by a Grant-in-Aid for Scientific Research (A) from the Japan Society for the Promotion of Science (JSPS) [15H02492 to T. Tomita]; Scientific Research on Innovative Areas from the Ministry of Education, Culture, Sports, Science, and Technology [26111705 to T. Tomita]; Strategic Research Program for Brain Sciences from the Japan Agency for Medical Research and Development (AMED) [JP16dm0107056 and JP17dm0107056 to T. Iwatsubo and T. Tomita]; the Daiichi Sankyo Foundation of Life Science [to T Tomita]; Ono Medical Research Foundation [to T. Tomita]; Takeda Science Foundation [to T. Tomita]; the Cell Science Research Foundation [to T. Tomita].

## Author contributions

KK and TTa designed and performed experiments, analyzed data, and wrote the manuscript. KI and MM designed and performed experiments regarding memantine, analyzed data, and discussed the results. NH, AK, and TIk analyzed the human brain and provided scientific inputs. TS and TCS provided *App* knockin mice and discussed the results. ST, YO, MA, TIw, and YH managed the project and provided scientific inputs throughout the study. TTo supervised the entire project, designed the experiments, analyzed data, and wrote the manuscript.

## Conflict of interest

K. I. and M. M. are full-time employees of Daiichi-Sankyo Co. Ltd., Tokyo, Japan. The authors have no additional financial interests.

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
