## [Review Process File · EMBO Molecular Medicine]

Loss of kallikrein-related peptidase 7 exacerbates amyloid pathology in Alzheimer's disease model mice

Kiwami Kidana, Takuya Tatebe, Kaori Ito, Norikazu Hara, Akiyoshi Kakita, Takashi Saito, Sho Takatori, Yasuyoshi Ouchi, Takeshi Ikeuchi, Mitsuhiro Makino, Takaomi C. Saïdo Masahiro Akishita, Takeshi Iwatsubo, Yukiko Hori, and Taisuke Tomita

Review timeline:

Submission date:	22 June 2017
Editorial Decision:	13 July 2017
Revision received:	29 October 2017
Editorial Decision:	10 November 2017
Revision received:	27 November 2017
Accepted:	07 December 2017

Editor: Céline Carret

Transaction Report:

1st Editorial Decision

13 July 2017

Thank you for the submission of your manuscript to EMBO Molecular Medicine. We have now heard back from the three referees whom we asked to evaluate your manuscript.

You will see below that the referees find the study interesting. Most comments are supportive and we feel that addressing them, experimentally when needed, would improve the data. Referee 3 has some concerns about the in vivo data as n is not always appropriate as well as the statistical analysis not optimum for multiple testing, and indeed strengthening this aspect of the work would be desirable.

We would welcome the submission of a revised version within three months, for further consideration and would like to encourage you to address all the criticisms raised as suggested to improve conclusiveness and clarity. Please note that EMBO Molecular Medicine strongly supports a single round of revision and that, as acceptance or rejection of the manuscript will depend on another round of review, your responses should be as complete as possible.

***** Reviewer's comments *****

Referee #1 (Remarks):

This is a very interesting paper describing for the very first time the in vivo effects of KLK7 on Amyloid metabolism. Thus this paper adds substantially to the previous publication by Shropshire et al. (2014), which only showed the in vitro capability of KLK7 to degrade synthetic Abeta.

The following points should be addressed in a revised version of this manuscript:

Abeta produced from another source (human cells or even better primary neurons from APPS1 mice) may be investigated as well, to prove that the observed effects are not specific to 7PA2

produced A β .

Is the 50% reduction of KLK7 mRNA in the Japanese AD brains also reflected on protein levels?

What means "Mock" in Fig. 3? Is this just a buffer control or an isotype control (what would be required)? Is the inhibitory activity of the antibody dose dependent?

In Fig. 3D numerous bands are observed. How can the authors be sure that they detect indeed the indicated dimers and trimers?

Why are the KLK7 driven effects in Fig. 3B so weak (hard to see). Was there a significant overexpression of KLK7?

Fig. 5D must be shown in a better resolution. In the overview I hardly see the plaques.

I do not understand Fig. 5E. What do the authors want to conclude from that Figure? Astrocytes surrounding amyloid plaques should occur in the presence and absence of KLK7. Is there overall a higher astrogliosis in KLK7 ko mice?

What about KLK7 protein levels in Fig. 7C?

Referee #2 (Comments on Novelty/Model System):

There are two novel findings reported. First, kallikrein-related peptidase 7 (KLK7) is shown to exacerbate amyloid pathology in transgenic mouse models of human amyloid pathology. Second, for the first time astrocytes are identified as source of A β -degrading proteolytic KLK7 and most important the NMDA antagonist memantine, a drug improved for the treatment of Alzheimer's disease is shown to induce KLK7 mRNA expression. Interestingly, the NMDA antagonist MK-801 has no effect on the regulation of astrocytic KLK7 suggesting that the subunit composition of astrocytic NMDA receptors differ from those of neurons since the latter bind MK-801. Regarding the technical quality, the animal models used throughout are crucial for what has been achieved. Both mouse models used throughout were most appropriate, the homozygous App-NL-G-F/NL-G-F mice that were produced by homologous recombination with a human App-NL-G-F gene that carries three familial mutations causing autosomal dominant Alzheimer's disease (AD) (FAD) and even so important and appropriate, the establishment and use of Klk7^{-/-} knockout mice and their crosses with homozygous App-NL-G-F mice.

Referee #2 (Remarks):

There are several minor points that should be considered and incorporated in the final manuscript. First and most important, the statement in the discussion that "increased expression and/or activity of KLK7 would provide beneficial effects for cognitively normal individuals with senile plaques" is not compatible with what has been published by Randy Bateman and coworkers (see for instance ANN NEUROL 2015;78:439-453). This group showed that in individuals with amyloid positivity A β 42 is selectively and rapidly removed from the CSF. About 50% of CSF A β 42, corresponding to \approx 30ng, is bound per hour to parenchymal A β 42 aggregates. These \approx 30ng correspond to \approx 5 of total A β 42 produced per hour, a 50% reduction in A β 42 by proteolysis mediated by KLK7 would not affect progression of AD pathology. For this at least 97.5% of A β 42 need to be removed (see the discussion in the aforementioned publication). Nevertheless, memantine has a great potential in combinatorial therapy, for instance in addition to immunotherapy or even as monotherapy after successful removal of amyloid by immunotherapy. Please amend.

Second, the first two sentences of the discussions need to be rewritten. As an example: "In this study, we identified KLK7 as a crucial astrocytic Since Klk7 deletion attenuates Thioflavin S-positive amyloid deposition, this indicates that astrocytes are involved in regulation of A β pathogenesis via the KLK 7 pathway. It was reported(Shropshire et al. 2014) but the pathophysiological role of KLK7 in the etiology of AD and the origin of KLK7 remained obscure."

Third, Figure 6D: LPS effect does not reach significance but it should be mentioned in the text that despite the current lack of significance a role in regulation of Klk7 cannot be excluded.

Fourth, Introduction: Change your misleading statement regarding "activation of remaining neurons to eliminate A β deposition" to what I would suggest: "remaining neurons by removal of A β deposits might not be suitable" ..

Fifth, use more appropriate terminology for the age of mice - replace "mice at 3-month-old" by "mice 3-months-old" or "mice 3 months of age"

Referee #3 (Comments on Novelty/Model System):

Too many experiments use insufficient n numbers in terms of mice to draw firm conclusions, and often statistical correction for multiple comparisons are not done.

Referee #3 (Remarks):

The manuscript by Kidana provides strong evidence that KLK7 is a novel Abeta degrading protease likely produced by astrocytes that can modulate amyloid deposition in vivo in the brain of mice. Evidence is provided that KLK7 expression may be decreased in the AD brain, but that in mice it increases as amyloid accumulates. These results are for the most part solid and well done though a few concerns are noted below. Some data on memantine is provided that suggests that this AD drug, which is used for cognitive enhancement in AD, could lower Abeta levels by increasing KLK7 expression. Though this data is intriguing, it is generated with 30 μ M Memantine and the effect size in vivo is quite small and based on a very small group size. A number of suggestions regarding this data are made below.

1. It would be highly useful to actually generate kinetic data (Km, Vmax etc) on the enzymology of ABeta degradation by KLK7. This data would help to understand its relative potential for turnover of Abeta by KLK7 compared to well studied abeta degrading enzymes such as NEP, IDE and ECE1/2.
2. Figure 3 in the 7PA2 cells it is not clear that all of the "oligomeric bands" are truly oligomers but represent novel ABeta immunoreactive cleavage products reported by Hass and colleagues.
3. The data on fibril degradation by KLK7 is important but buried in supplemental information. I think this should be in the main part of the manuscript.
4. Figure 5. The mouse data looks convincing in terms of effect size but is poorly documented. An n of 3-4 is simply insufficient to draw firm conclusions. This data needs to be repeated with a larger group size.
5. Figure 6. A t-test is not appropriate as multiple comparisons are being made the data needs to be adjusted for multiple comparisons.
6. Both the human KLK7 expression data and the mouse data could be greatly improved by mining public data sets now available at SAGE AD-AMP portal. This data is freely accessible and contains hundreds of samples from AD and control brain and longitudinal data from APP and tau mice.
7. Memantine effects. What is the rationale for using 30 μ M? A dose response should be shown on primary astrocytes. The expression data in vivo in figure 7c is based on too few of an n number, and again is not adjusted for multiple comparisons. The effect size is marginal. I don't think this data is robust enough to publish in its current form. Toxicity controls are lacking and it is not clear that the mechanism here is worked out. Again mining public data for memantine effects on Abeta levels in CSF would be a logical implication of this data that could be rapidly done through ADNI. Unfortunately the most relevant preclinical experiment would be to look at whether memantine lowers Abeta in an APP mouse crossed into a KLK7 background.
8. The data on tau again shows a modest effect size and is based on too low of an n number.

1st Revision - authors' response

29 October 2017

RESPONSE TO REVIEWER 1:

Comment 1: *Abeta produced from another source (human cells or even better primary neurons from APPPS1 mice) may be investigated as well, to prove that the observed effects are not specific to 7PA2 produced Abeta.*

Response: We thank the reviewer for this pertinent comment. In accordance with the Reviewer's comment, we examined the A β degrading activity of CCF-STTG1 cells using the conditioned medium of BE(2)-C cells, which is derived from Human Caucasian neuroblastoma and secrete the substantial amount of endogenous A β . We confirmed that human A β from BE(2)-C cells was also degraded by the conditioned medium of CCF-STTG1 cells in a similar manner to that from 7PA2 cells (Supplementary Fig. 1C).

Comment 2: *Is the 50% reduction of KLK7 mRNA in the Japanese AD brains also reflected on protein levels?*

Response: We validated several commercially available KLK7 antibodies as well as custom-made antibodies using lysates from *Klk7*^{-/-} mouse. However, no antibody specifically detected an endogenous KLK7 protein in the brain lysates *Klk7* wt, hets and KO mouse so far (see left). We have been trying to generate antibodies that detect the endogenous KLK7 in brain, and will report the characters of these antibodies elsewhere in future.

Figure: Immunoblot analysis using rabbit anti-kallikrein 7 (Abcam #ab28309). Although *Klk7*-like bands (28~30 kDa, arrowheads) were detected, these proteins were unchanged in KO mouse lysate.

Comment 3: *What means "Mock" in Fig. 3? Is this just a buffer control or an isotype control (what would be required)? Is the inhibitory activity of the antibody dose dependent?*

Response: We appreciate the reviewer for critical reading. We used PBS (i.e., a buffer control) as Mock in Figure 3A and 4B. To address the concern by the reviewer, we examined the effect of anti-V5 Tag (R960-25, Thermo Fisher Scientific) and anti-LR11 (#611860, BD Biosciences) monoclonal antibodies as an isotype control of MAB2624 (mouse IgG2a) and observed the same results as a buffer control in both the conditioned medium of CCF-STTG1 and primary astrocytes culture (Supplementary Fig. 3A and 4B). In addition, we also confirmed the dose dependence of MAB2624 in the conditioned medium of CCF-STTG1 (Fig. 3B).

Comment 4: *In Fig. 3D numerous bands are observed. How can the authors be sure that they detect indeed the indicated dimers and trimers?*

Response: We understand the reviewer's concern. We tested the effects of Congo Red and γ -secretase inhibitor DAPT on the secreted A β species from 7PA2 cells (See right). We utilized 6E10 antibody, which reacts with human A β species starting at Asp1, but not with A β x- peptides in this assay. We found that the bands between 2 to 10 kDa were completely diminished by γ -secretase inhibitor DAPT. As A η is generated independently of γ -secretase, these bands represent an A β -related protein including A β monomer at 4 kDa. Congo Red decreased the bands between 5 to 10 kDa, suggesting that these bands represent oligomeric form of A β . However, as we have not analyzed the biochemical properties of these bands, we referred them as "oligomer" A β .

Comment 5: *Why are the KLK7 driven effects in Fig. 3B so weak (hard to see). Was there a significant overexpression of KLK7?*

Response: We agree with the reviewer. One possibility is that the protein concentration of overexpressed KLK7 used in this assay was low. In fact, relatively high protein concentration (i.e., μ g/mL order) is required for complete degradation of A β by recombinant KLK7 (Supplementary

Fig. 3). In addition, we shortened the incubation time for 3 hours in this experiment to make the difference clearer.

Comment 6: *Fig. 5D must be shown in a better resolution. In the overview I hardly see the plaques.*

Response: We thank the reviewer for the comment. We have changed immunohistochemical image of the Figure 5D.

Comment 7: *I do not understand Fig. 5E. What do the authors want to conclude from that Figure? Astrocytes surrounding amyloid plaques should occur in the presence and absence of KLK7. Is there overall a higher astrogliosis in KLK7 ko mice?*

Response: We appreciate the reviewer for critical reading. We observed the significant increase of GFAP-positive activated astrocytes around plaques in *App*^{NL-G-F/NL-G-F}; *Klk7*^{-/-} mouse, presumably accelerated maturation of amyloid plaques at this age. We have replaced Figure 5E to indicate that astrocyte was not so activated, even around Thioflavin S positive plaques in *App*^{NL-G-F/NL-G-F}; *Klk7*^{+/+} mouse.

Comment 8: *What about KLK7 protein levels in Fig. 7C?*

Response: Please see the answer for comment 2.

RESPONSE TO REVIEWER 2:

Comment 1: *First and most important, the statement in the discussion that "increased expression and/or activity of KLK7 would provide beneficial effects for cognitively normal individuals with senile plaques" is not compatible with what has been published by Randy Bateman and coworkers (see for instance ANN NEUROL 2015;78:439-453). This group showed that in individuals with amyloid positivity Aβ42 is selectively and rapidly removed from the CSF. About 50% of CSF Aβ42, corresponding to ≈30ng, is bound per hour to parenchymal Aβ42 aggregates. These ≈30ng correspond to ≈5 of total Aβ42 produced per hour, a 50% reduction in Aβ42 by proteolysis mediated by KLK7 would not affect progression of AD pathology. For this at least 97.5% of Aβ42 need to be removed (see the discussion in the aforementioned publication). Nevertheless, memantine has a great potential in combinatorial therapy, for instance in addition to immunotherapy or even as monotherapy after successful removal of amyloid by immunotherapy. Please amend.*

Response: We appreciate the precise comment by the reviewer. Importantly, KLK7 degrades not only Aβ monomer, but oligomer and fibrils (Fig. 3). Thus, we expected that the clearance of aggregated Aβ species as senile plaques is also facilitated by the activation of KLK7. But as we have not examined the kinetics of amyloid plaques in memantine-treated nor *Klk7* knockout mouse brain, we cannot hypothesize the beneficial effect on the individuals with senile plaques. Also, we agree with the potential of memantine in the combination therapy with the immunotherapy. We deleted the indicated sentences, added phrases regarding possible combination effect of memantine in the discussion section.

Comment 2: *Second, the first two sentences of the discussions need to be rewritten. As an example: "In this study, we identified KLK7 as a crucial astrocytic Since *Klk7* deletion attenuates Thioflavin S-positive amyloid deposition, this indicates that astrocytes are involved in regulation of Aβ pathogenesis via the KLK 7 pathway. It was reported(Shropshire et al. 2014) ,but the pathophysiological role of KLK7 in the etiology of AD and the origin of KLK7 remained obscure."*

Response: We appreciate the comment by the reviewer. We amended the sentences.

Comment 3: *Figure 6D: LPS effect does not reach significance but it should be mentioned in the text that despite the current lack of significance a role in regulation of *Klk7* cannot be excluded.*

Response: We appreciate the reviewer for critical reading. We added following sentence in the result section; "Although we are unable to exclude the possibility that lipopolysaccharide has some regulatory role in the *Klk7*,"

Comment 4: *Fourth, Introduction: Change your misleading statement regarding "activation of remaining neurons to eliminate Aβ deposition" to what I would suggest: "remaining neurons by removal of Aβ deposits might not be suitable".*

Response: We appreciate the reviewer for critical reading. We changed the text in the Introduction.

Comment 5: *Fifth, use more appropriate terminology for the age of mice - replace "mice at 3-month-old" by "mice 3-months-old" or "mice 3 months of age".*

Response: In accordance with the Reviewer's comment, we unified the writing from "month-old" to "months of age".

RESPONSE TO REVIEWER 3:

Comment 1: *It would be highly useful to actually generate kinetic data (Km, Vmax etc) on the enzymology of ABeta degradation by KLK7. This data would help to understand its relative potential for turnover of Abeta by KLK7 compared to wells studied abeta degrading enzymes such as NEP, IDE and ECE1/2.*

Response: We appreciate the comment by reviewer. We have investigated kinetic data of recombinant KLK7 protein on the degradation of synthetic A β . However, our results lacked a sufficient reliability because of relatively low proteolytic activity (KLK7 requires 12~24 hours for complete degradation of A β in our assay). Previous report indicates that Kcat/KM value of KLK7 for FRET-based A β (16-23) peptide is 0.24 ± 0.05 $\mu\text{M}/\text{min}$ (Shropshire et al., Biol Chem 2014). Kcat/KM values for neprilysin and IDE for A β (1-7) were 0.21 and 0.91 $\mu\text{M}/\text{min}$, respectively (Chen et al., J Neurosci Methods. 2010). These data suggest that these enzymes cleave A β at similar level, although these *in vitro* experiments utilized different peptide substrate. Thus, it is very difficult to compare the enzymatic properties of these enzymes. Moreover, neprilysin, IDE and ECE are able to cleave A β at several positions. In contrast, KLK7 cleaves at single site at hydrophobic core region in aggregated A β . Thus, we hypothesize that KLK7 plays an important role to disaggregate the oligomeric forms of A β by cutting at hydrophobic core, then remaining N- and C-terminal halves of A β are degraded by the other enzymes.

Comment 2: *Figure 3 in the 7PA2 cells it is not clear that all of the "oligomeric bands" are truly oligomers but represent novel ABeta immunoreactive cleavage products reported by Hass and colleagues.*

Response: We appreciate the comment by the reviewer. Please see our response to the comment 4 by the reviewer 1.

Comment 3: *The data on fibril degradation by KLK7 is important but buried in supplemental information. I think this should be in the main part of the manuscript.*

Response: We appreciate kind suggestion by the reviewer. We added the data regarding fibril degradation in Fig. 3 from the supplementary data.

Comment 4: *Figure 5. The mouse data looks convincing in terms of effect size but is poorly documented. An n of 3-4 is simply insufficient to draw firm conclusion. This data needs to be repeated with a larger group size.*

Response: We understand the reviewer's concern. In this revision, we were able to add samples in the analysis of tau phosphorylation (see the answer for comment 8). However, due to the limitation of the animal facility and time, because we could not prepare additional *App*^{NL-G-F/NL-G-F}; *klk7*^{-/-} mice with appropriate age in this study. Nevertheless, we are planning to analyze congenic mice at older age and report in the future.

Comment 5: *A t-test is not appropriate as multiple comparison are being made the data needs to be adjusted for multiple comparisons.*

Response: We appreciate the reviewer for critical reading. We conducted the statistical analysis of Figure 6C, Figure 7D and Figure 7E using Tukey's test.

Comment 6: *Both the human KLK7 expression data and the mouse data could be greatly improved by mining public data sets now available at SAGE AD-AMP portal. This data is freely accessible and contains hundreds of samples form AD and control brain and longitudinal data from APP and tau mice.*

Response: We appreciate kind suggestion by the reviewer. We analyzed the expression data of human brain samples using both Mayo RNAseq and Mount Sinai Brain Bank (MSBB) AD cohorts. In Mayo RNAseq database, we observed that the KLK7 expression was significantly reduced in the temporal cortex of AD group compared with that of control ($q < 0.05$). In MBSS RNA-seq database, we found that the expression levels of KLK7 mRNA were significantly reduced in the BM22 (Superior temporal gyrus) and BM36 (Parahippocampal gyrus) with the progression of Braak NFT

stage as well as plaque load. Unfortunately the expression levels of *Klk7* mRNA in the database of MAPT_P301L, rTG4510, APPS1 and TgCRND8 of AD model mice were quite low. Thus, we were unable to perform appropriate analyses. Nevertheless, we have obtained similar results from different cohorts, supporting our notion that *KLK7* mRNA expression is significantly reduced in the brains of AD patients.

Comment 7: *Memantine effects. What is the rationale for using 30 μ M? A dose response should be shown on primary astrocytes. The expression data in vivo in figure 7c is based on too few of an n number, and again is not adjusted for multiple comparisons. The effect size is marginal. I don't think this data is robust enough to publish in its current form. Toxicity controls are lacking and it is not clear that the mechanism here is worked out. Again mining public data for memantine effects on Abeta levels in CSF would be a logical implication of this data that could be rapidly done through ADNI. Unfortunately the most relevant preclinical experiment would be to look at whether memantine lowers Abeta in an APP mouse crossed into a *KLK7* background.*

Response: We understand the reviewer's concerns. First, we examined the cytotoxicity of memantine against primary astrocytes using alamar blue test and decided to use memantine up to 30 μ M. Next, we examined the A β degrading activity of the conditioned medium of primary astrocytes treated with memantine, and confirmed that the A β degrading activity was augmented in a dose-dependent manner (Supplementary Fig. 10).

Regarding ADNI sample, we expect that there is no difference in the level of A β level in cerebrospinal fluid between memantine recipients and non-recipients of US-ADNI subjects by following reason. In general, A β 42 level in CSF of AD patients is significantly decreased (Shaw et al., Ann Neurol 2009). Thus, it would be difficult to discriminate whether the additional decrease of CSF A β 42 by upregulation of *KLK7* with memantine or the progression of AD pathology. In addition, CSF A β 40 level has not examined in ADNI study. Nevertheless, we are willing to examine the CSF A β 42 level in cerebrospinal fluid from the patients with or without memantine in the future.

Comment 8: *The data on tau again shows a modest effect size and is based onto low of n number.*

Response: We understand the reviewer's concern. Here we added the number of samples to 4-5 and analyzed by Tukey's test (Supplementary Figure 8B). Again, we confirmed the increase of phosphorylated tau in *App^{NL-G-F/NL-G-F}; Klk7^{-/-}*. We also observed the relationship between *KLK7* expression and the NFT stage from analyses by MSBB RNA-seq. Thus, we believe the loss of *KLK7* expression accelerates the amyloid deposition, thereby increasing the tau phosphorylation at this age. However, to further clarify the role of *KLK7* in tau pathology, additional studies using aged mice would be needed. We will report this issue in the future.

2nd Editorial Decision

10 November 2017

Thank you for the submission of your revised manuscript to EMBO Molecular Medicine. We have now received the enclosed reports from the referees that were asked to re-assess it. As you will see the reviewers are now globally supportive and I am pleased to inform you that we will be able to accept your manuscript pending the following final amendments:

1) Please address the changes commented and recommended by the referees. Please provide a letter INCLUDING the reviewer's reports and your detailed responses to their comments (as Word file).

***** Reviewer's comments *****

Referee #1 (Remarks for Author):

Abeta produced from another source (human cells or even better primary neurons from APPS1 mice) may be investigated as well, to prove that the observed effects are not specific to 7PA2 produced Abeta.

The authors have addressed this point appropriately by adding the requested new data.

Is the 50% reduction of *KLK7* mRNA in the Japanese AD brains also reflected on protein levels? The authors tried, but there is no antibody available, which recognizes endogenous *KLK7*. Thus, the

authors have addressed this point appropriately.

What means "Mock" in Fig. 3? Is this just a buffer control or an isotype control (what would be required)? Is the inhibitory activity of the antibody dose dependent?

The authors have addressed this point appropriately by adding the requested new data.

In Fig. 3D numerous bands are observed. How can the authors be sure that they detect indeed the indicated dimers and trimers?

I am still not very happy with this figure. Now the bands are labeled as oligomers (previously dimer and trimer). Furthermore in the figure added to comment 4, Congo Red reduced Abeta generation. This is surprising as the original paper (Podlisny et al., JBC 1995) showed a dramatic increase. I understand that the authors want to prove that KLK7 also degrades oligomers, which are believed to be toxic. However, at the end the authors would need to check if KLK7 is capable to block the LTP lowering activity of purified and characterized oligomers. I would therefore recommend to remove Fig. 3F.

Why are the KLK7 driven effects in Fig. 3B so weak (hard to see). Was there a significant overexpression of KLK7?

This has been addressed as good as possible.

Fig. 5D must be shown in a better resolution. In the overview I hardly see the plaques.

This has been addressed by exchanging the figure.

I do not understand Fig. 5E. What do the authors want to conclude from that Figure? Astrocytes surrounding amyloid plaques should occur in the presence and absence of KLK7. Is there overall a higher astrogliosis in KLK7 ko mice?

I think the authors want to say that the increased amyloid burden in the absence of KLK7 increases neuritic and astrocytic pathology. Why not making a separate figure, which includes the data from the Supplementary Fig. 8 and an appropriate description? This would allow the authors to draw a nice conclusion!

What about KLK7 protein levels in Fig. 7C?

This has been described as described in the response to comment 2.

Referee #3 (Remarks for Author):

The manuscript is improved. One area of lingering concern is the kinetic data is still a gap in the current manuscript. Some discussion of this should be included as to the relative effect this protease has compared to others.

2nd Revision - authors' response

27 November 2017

Referee #1 (Remarks for Author):

Abeta produced from another source (human cells or even better primary neurons from APPPS1 mice) may be investigated as well, to prove that the observed effects are not specific to 7PA2 produced Abeta.

The authors have addressed this point appropriately by adding the requested new data.

We appreciate the reviewer's comment.

Is the 50% reduction of KLK7 mRNA in the Japanese AD brains also reflected on protein levels?

The authors tried, but there is no antibody available, which recognizes endogenous KLK7. Thus, the authors have addressed this point appropriately.

We appreciate the reviewer's comment.

What means "Mock" in Fig. 3? Is this just a buffer control or an isotype control (what would be

*required)? Is the inhibitory activity of the antibody dose dependent?
The authors have addressed this point appropriately by adding the requested new data.*

We appreciate the reviewer's comment.

*In Fig. 3D numerous bands are observed. How can the authors be sure that they detect indeed the indicated dimers and trimers?
I am still not very happy with this figure. Now the bands are labeled as oligomers (previously dimer and trimer). Furthermore in the figure added to comment 4, Congo Red reduced Abeta generation. This is surprising as the original paper (Podlisny et al., JBC 1995) showed a dramatic increase. I understand that the authors want to prove that KLK7 also degrades oligomers, which are believed to be toxic. However, at the end the authors would need to check if KLK7 is capable to block the LTP lowering activity of purified and characterized oligomers. I would therefore recommend to remove Fig. 3F.*

We understand the reviewer's concern. We have deleted the result regarding oligomeric Abeta (Fig. 3F) and related discussions.

*Why are the KLK7 driven effects in Fig. 3B so weak (hard to see). Was there a significant overexpression of KLK7?
This has been addressed as good as possible.*

We appreciate the reviewer's comment.

*Fig. 5D must be shown in a better resolution. In the overview I hardly see the plaques.
This has been addressed by exchanging the figure.*

We appreciate the reviewer's comment.

*I do not understand Fig. 5E. What do the authors want to conclude from that Figure? Astrocytes surrounding amyloid plaques should occur in the presence and absence of KLK7. Is there overall a higher astrogliosis in KLK7 ko mice?
I think the authors want to say that the increased amyloid burden in the absence of KLK7 increases neuritic and astrocytic pathology. Why not making a separate figure, which includes the data from the Supplementary Fig. 8 and an appropriate description? This would allow the authors to draw a nice conclusion!*

We thank the reviewer for the comment. We moved the Fig. 5E to Appendix Fig. S8D. Also, we have separated the description regarding amyloid-induced pathology (i.e., phospho tau, neuritic change and astrogliosis) in the result section.

*What about KLK7 protein levels in Fig. 7C?
This has been described as described in the response to comment 2.*

We appreciate the reviewer's comment.

Referee #3 (Remarks for Author):

The manuscript is improved. One area of lingering concern is the kinetic data is still a gap in the current manuscript. Some discussion of this should be included as to the relative effect this protease has compared to others.

We understand the reviewer's concern. We added the comment on the effects of known Abeta degrading enzymes on Abeta metabolism in mouse brain at discussion section.

Corresponding Author Name: Taisuke Tomita

Manuscript Number: EMM-2017-08184-V2